# Reducing the Invasiveness of Low- and High-Grade Endometrial Cancers in Both Primary Human Cancer Biopsies and Cell Lines by the Inhibition of Aquaporin-1 Channels

**DOI:** 10.3390/cancers15184507

**Published:** 2023-09-11

**Authors:** Sidra Khan, Noor A. Lokman, Martin K. Oehler, Carmela Ricciardelli, Andrea J. Yool

**Affiliations:** 1School of Biomedicine, University of Adelaide, Adelaide, SA 5000, Australia; sidra.khan@adelaide.edu.au; 2Adelaide Medical School, Robinson Research Institute, University of Adelaide, Adelaide, SA 5000, Australia; noor.lokman@adelaide.edu.au (N.A.L.); oehler.mk@gmail.com (M.K.O.); 3Department of Gynaecological Oncology, Royal Adelaide Hospital, Adelaide, SA 5000, Australia

**Keywords:** AQP, cation channel, cell invasion, reproductive cancer, gynecological oncology, estrogen, progesterone, AqB011, 5-PMFC

## Abstract

**Simple Summary:**

A major clinical concern for patients with endometrial cancer is the aggressive spread of cancer cells into other tissue regions, worsening symptoms, eluding medical therapy, disrupting organ functions, and increasing risk of relapse. As reported here, drug-like agents that we have found block the ion channel function of a specific membrane protein known as Aquaporin-1 (AQP1) can be used to inhibit invasiveness of endometrial cancer cells. In vitro invasion assays were done with cell lines originally established from endometrial cancers, in parallel with assays on primary endometrial cancer cells isolated from endometrial cancer tissue at the time of surgery. The exciting discovery was that the AQP1 ion channel blockers inhibited invasiveness of both low- and high-grade endometrial cancer types in established cell lines and primary cancer cells at doses that did not cause toxic side effects. Low toxicity is encouraging for potential co-administration with other first-line treatments. Pharmacological agents optimized from lead compounds could be invaluable for slowing the progression of both low- and high grade cancers, including a challenging subset which are unresponsive to current hormone-based therapies. Further work on refining structures of the drugs, their distribution and clearance will be needed for progress towards bench-to-bedside clinical translation.

**Abstract:**

Aquaporin (AQP) channels in endometrial cancer (EC) cells are of interest as pharmacological targets to reduce tumor progression. A panel of compounds, including AQP1 ion channel inhibitors (AqB011 and 5-(phenoxymethyl) furan-2-carbaldehyde, PMFC), were used to test the hypothesis that inhibition of key AQPs can limit the invasiveness of low- and high-grade EC cells. We evaluated the effects on transwell migration in EC cell lines (Ishikawa, MFE-280) and primary EC cells established from surgical tissues (*n* = 8). Quantitative PCR uncovered classes of *AQPs* not previously reported in EC that are differentially regulated by hormonal signaling. With estradiol, Ishikawa showed increased *AQPs 5*, *11*, *12*, and decreased *AQPs 0* and *4*; MFE-280 showed increased *AQPs 0*, *1*, *3*, *4*, *8*, and decreased *AQP11*. Protein expression was confirmed by Western blot and immunocytochemistry. AQPs 1, 4, and 11 were colocalized with plasma membrane marker; AQP8 was intracellular in Ishikawa and not detectable in MFE-280. AQP1 ion channel inhibitors (AqB011; PMFC) reduced invasiveness of EC cell lines in transwell chamber and spheroid dispersal assays. In Ishikawa cells, transwell invasiveness was reduced ~41% by 80 µM AqB011 and ~55% by 0.5 mM 5-PMFC. In MFE-280, 5-PMFC inhibited invasion by ~77%. In contrast, proposed inhibitors of AQP water pores (acetazolamide, ginsenoside, KeenMind, TGN-020, IMD-0354) were not effective. Treatments of cultured primary EC cells with AqB011 or PMFC significantly reduced the invasiveness of both low- and high-grade primary EC cells in transwell chambers. We confirmed the tumors expressed moderate to high levels of AQP1 detected by immunohistochemistry, whereas expression levels of AQP4, AQP8, and AQP11 were substantially lower. The anti-invasive potency of AqB011 treatment for EC tumor tissues showed a positive linear correlation with AQP1 expression levels. In summary, AQP1 ion channels are important for motility in both low- and high-grade EC subtypes. Inhibition of AQP1 is a promising strategy to inhibit EC invasiveness and improve patient outcomes.

## 1. Introduction

Endometrial cancer (EC) is the most common gynecologic malignancy, affecting, on average, 19 out of every 100,000 women in the United States [1,2]. Postmenopausal onset of EC accounts for approximately 75–80% of cases [3]; however, an additional 15–20% of women experience EC before the age of 50 [4]. A major complication in all cases is the aggressive spread of EC cells into other tissues, escaping therapies and increasing the risk of recurrence. Pharmacological inhibitors of cancer invasiveness combined with current therapies could improve treatment outcomes [5]. One of the candidate protein classes for therapeutic targeting are the aquaporins (AQPs), which, when upregulated in diverse cancers, have been shown to facilitate motility and metastatic spread [6]. Pharmacological small molecule AQP blockers have been used to inhibit cancer cell motility [7], with promising results in models of colon, breast, prostate, and ovarian cancers, both in vitro and in vivo [8,9,10,11,12]. However, the value of AQP pharmacological inhibitors in controlling EC invasiveness has not been previously investigated.

The thirteen classes of aquaporins (AQPs 0 to 12) expressed in the human show distinct tissue-specific distributions and diverse physiological functions, including maintaining fluid balance in the brain and eyes; concentrating urine; regulating transepithelial hydration; modulating cardiovascular and lung functions; secreting cerebrospinal fluid, tears, and saliva; and more [13,14,15]. ECs were shown previously to express multiple classes of aquaporins (AQPs 1, 2, 3, and 5) [9,16,17,18]. Genetic knockdown of *AQPs 2* and *5* was found to reduce motility of EC cells in vitro [9,17]. However, a comprehensive evaluation of the full set of human *AQPs* 0 to 12 and their regulation by steroid hormone signaling in EC remains lacking. AQP channels facilitate fluxes across cell membranes of water, glycerol, and other solutes, including ions in some AQP subtypes [13], including AQP1, which conducts monovalent cations through the tetrameric central pore of the channel [19,20]. Previously unknown classes of AQPs in EC cell lines now identified here are *AQPs 0*, *4*, *11*, *8*, and *12A*. AQP drug discovery research is still at an early stage, in that there is not a comprehensive set of agents for all human AQP channels; however, the portfolio continues to be expanded [7], and available agents were tested here as a starting point for further work.

The main aim of this study was to assess the effects of pharmacological inhibition of aquaporin channels on cell invasiveness in established EC cell lines and primary EC cells established from human EC tumor tissues following surgery. A library of pharmacological modulators of aquaporins was tested for effectiveness in controlling invasiveness in EC cells. The primary outcome of this work was the novel demonstration that AQP1 ion channel inhibitors restrained the invasiveness of EC cell lines and primary cancer cells from both low- and high-grade EC grades. AQP1 expression was confirmed by quantitative PCR and immunohistochemistry of fixed paraffin-embedded cancer tissues. 

We also showed that the ovarian hormones estradiol and progesterone regulate the expression of key classes of AQPs. Hypertrophy of endometrium in EC occurs in response to estrogen overexposure secondary to conditions such as early menarche, late menopause, obesity, infertility, unopposed estrogen-containing hormone replacement therapy, and metabolic syndrome [21]. When diagnosed and treated in the early stages, EC prognosis is often favorable; however, standard treatments involving removal of the uterus and ovaries carry major consequences, including loss of fertility and early onset of menopausal symptoms, especially in young women [22]. Other treatment options (chemotherapy, radiation therapy, and progestin therapy) provide clinical benefits but also have adverse side effects [23] and are associated with high recurrence rates [24]. The results here indicate that pharmacological agents targeting AQP1 to inhibit invasiveness could be used as a complement to conventional therapy in the clinical management of EC, pending confirmation in vivo. 

## 2. Materials and Methods

### 2.1. Pharmacological Agents

A summary of the agents, structures, and doses used in these experiments is provided in Table 1. Acetazolamide (A), used clinically as a carbonic anhydrase inhibitor and diuretic, has been proposed to also block water channel activity of AQPs 1 [25] and 4 [26], though efficacy varies between experimental systems [7]. Ginsenoside (Rg3) (B), derived from the medicinal plant Panax ginseng, has diverse biological effects, including proposed effects as a water pore inhibitor for AQP1 [8,10]. KeenMind (lot K0078702), a dietary supplement commercially produced from the dried medicinal herb *Bacopa monnieri*, has been shown to block water and H_2_O_2_ permeation through the intrasubunit pore of AQP1, with activity primarily attributed to the major component Bacopaside II [27]. AqB011, a synthetic derivative of the diuretic agent bumetanide, has been shown to interact with the gating domain of AQP1 and block the ion conductance without impairing water channel activity [11,28]. Also shown to act as selective AQP1 ion channel blockers are the furan compounds, 5-hydroxymethyl-2-furfural (5-HMF) and 5 phenoxymethylfuran-2-carbaldehyde (5-PMFC), which, similarly, do not alter water flux through the parallel intrasubunit pores of the AQP1 tetramer [29,30]. IMD-0354 (N-(3, 5-bis-trifluoromethyl-phenyl)-5-chloro-2-hydroxy-benzamide) is suggested to interfere with the water channel activity of AQP4 by direct blocking or via downregulation of expression [13]. TGN-020 (n-(1,3,4-thiadiazol-2-yl)nicotinamide) has been proposed to act as a water channel blocker for AQP4 [31,32]. The isolated medicinal herb products curcumin (from turmeric) and resveratrol (from grape seed) have diverse biological effects on growth factor and interleukin signaling pathways, and they have suggested roles in decreasing activity of AQP3 and AQP4 by downregulation of expression [33,34,35]. All pharmacological agents were purchased from Merck (formerly Sigma Aldrich Chemical Company; Melbourne, VIC, Australia), except KeenMind, which was purchased from Terry White Chemist (Adelaide, SA, Australia), and AqB011, which was custom synthesized by Dr G Flynn (SpaceFill LLC, Boseman, MT, USA).

### 2.2. Human EC Biopsy Collection and Primary Cell Culture

Tumor tissues were collected after hysterectomy from EC patient donors with informed consent and protocols approved by the Royal Adelaide Hospital Human Ethics Committee (CALHN Ref # 16700). Methods for the isolation of primary cancer cells from tissue biopsies have been published previously [36]. The cancer tissues were each divided into 3–5 pieces; one piece was fixed in 10% formalin (Chem Supply Pty Ltd., Port Adelaide, SA, Australia) for histology, and the remaining tissue was cryopreserved in liquid nitrogen and used to establish primary cultures of EC cells. For primary cultures, 0.5 to 1 g of tissue was cut into small pieces (1 to 2 mm^3^), transferred into 2.4 U/mL Dispase II (cat # 17105-041; ThermoFisher Scientific; Waltham, MA, USA) in Advanced Dulbecco’s Modified Eagle’s Medium (DMEM/F12) (ThermoFisher Scientific, cat. # 12634-010; Waltham, MA, USA), and incubated for 1 h at 37 °C in a shaking incubator at 200 rpm. Samples were passed through 70 µm cell strainers (pluriSelect, 43-50070-51; Leipzig, Germany) under pressure from a 10 mL syringe into 50 mL tubes (Corning 339652; Merck, Macquarie Park, NSW, Australia). Tubes were centrifuged at 1500 rpm; cell pellets were re-suspended in advanced DMEM/F12 supplemented with 10% fetal bovine serum (FBS) (Gibco; 10270106; Waltham, MA, USA), 100 U/mL penicillin, 100 µg/mL streptomycin sulfate (ThermoFisher Scientific, Ref: 15140122; Waltham, MA, USA), 100 µg/mL amphotericin B (Merck; 1397-89-3; Macquarie Park, NSW, Australia), and 2 mM GlutaMAX (ThermoFisher Scientific; #35050061; Waltham, MA, USA). Cells were grown as monolayers and used at passage 0 or 1 to minimize phenotypic variation and fibroblast contamination. The clinicopathological information of the patients used to isolate the primary cells is summarized in Appendix A. The epithelial nature of the primary cancer cells was confirmed using pan-cytokeratin staining [37]. In this study, a threshold of at least 60% pancytokeratin-positive cells per field of view in EC samples was set as a criterion for inclusion. Examples of pan-cytokeratin immunostaining in low- and high-EC primary cells are shown in Appendix A. The morphology of EC cells in primary cultures showed rounded or short spindle-shaped appearances, distinctly different from the elongated, flattened, and densely aggregated pattern in fibroblast cells (Appendix A).

### 2.3. EC Cell Line Cultures and Steroid Hormone Treatments

Two EC cell lines, Ishikawa 3-H-12-Luc (JCRB1579, Ishikawa, purchased January 2019 at passage 4) and MFE-280 (ECACC 98050131, purchased January 2021 at passage 68), were obtained from CellBank Australia (Cancer Institute, NSW, Australia). Additional cell passages were held at less than 10 for all experimental work. Cell lines were used within 6 months after cryopreservation. Negative status for mycoplasma was confirmed using MycoStrip test kits (InvivoGen, San Diego, CA, USA). The Ishikawa line was maintained in DMEM supplemented with 10% FBS, 100 U/mL penicillin, 100 µg/mL streptomycin, and 2 mM GlutaMAX. The MFE-280 line was maintained in DMEM/F12 supplemented with 10% FBS, penicillin, streptomycin, and 0.005 mg/mL insulin. DMEM, FBS, GlutaMAX, penicillin–streptomycin, insulin (I6634-25 mg), 17β-Estradiol (E8875-250 mg), and progesterone (P0130-25G) were purchased from Merck (Macquarie Park, NSW, Australia). At 24 h before steroid hormone treatments, culture medium was changed to phenol red-free DMEM/F12 (ThermoFisher Scientific; 21041025; Waltham, MA, USA) supplemented with 10% heat-inactivated charcoal-stripped (CS)-FBS to rule out estrogenic effects of FBS [38]. Ishikawa and MFE-280 cell lines correspond to FIGO (International Federation of Gynecology and Obstetrics; https://www.figo.org/) endometroid cancers grades 1 and 3, respectively. Ishikawa is an estrogen and progesterone receptor-positive cell line originally established from endometroid cancer from a 39-year-old pre-menopausal woman [39]. A FIGO grade 1 classification for Ishikawa is based on the *PTEN* mutation marker (database Expasy Cellosaurus; CVCL_D199) and TCGA classification as Microsatellite Instability High (MSI-H) type [40]. The MFE-280 cell line, originally derived from a recurrent, poorly differentiated endometrial carcinoma from a 78-year-old post-menopausal patient [41], is classified as FIGO grade 3 based on TP53 gene mutations (CVCL_1405) and TCGA designation as Copy Number High (CNH) type [40].

### 2.4. Immunohistochemistry for Human EC Biopsy Tissues

Paraffin-embedded tissue sections were prepared using published protocols [42]. Briefly, tissue sections mounted on coated slides (Trajan; # 471042221) were incubated on a hot plate at 60 °C for 2 h followed by dewaxing in xylene and ethanol. After PBS washes, slides were incubated with 0.3% hydrogen peroxide (H_2_O_2_) for 5 min to quench endogenous peroxidase activity. Antigen retrieval was performed with citric acid buffer (10 mM, pH 6) in a steam microwave (Sixth Sense, Whirlpool, VIC, Australia) at 100 °C for 10 min. Tissues were incubated with 5% goat serum for 30 min followed by incubation with primary antibodies (rabbit anti-AQP1, 1:750 dilution (Merck Macquarie Park NSW Australia), A5560, anti-hAQP4 (Merck, #HPA014784-100UL, 0.5 µg/mL, used at 1:500); anti-hAQP8 (Merck HPA046259, 2 µg/mL, used at 1:250); and anti-hAQP11 (Abcam; ab122821, 1 µg/mL, used at 1:100; Melbourn, VIC, Australia) at 4 °C overnight. Subsequently, tissue sections were incubated sequentially with biotinylated goat anti-rabbit (at 1:400, Dako; E0433), followed by streptavidin-HRP conjugated antibody (at 1:500, Dako; P0397) for 1 h at room temperature. Immunoreactivity was detected using diaminobenzidine/H_2_O_2_ substrate (Merck). Sections were counterstained with 10% hematoxylin (Merck), mounted in Pertex (Medite Medizintechnik, Burgdorf Niedersachsen, Germany), and digitally scanned using the NanoZoomer Digital Pathology system (Hamamatsu Photonics K.K.; Shizuoka, Japan). QuPath software (0.3.2, Ireland, https://qupath.readthedocs.io/en/0.4/ (accessed 2022)) was used to measure the AQP immunostaining intensity and positivity using the H-score [43]. The H-score typically ranges from 0 for “no signal” to 300 for a “strong signal”. In this analysis, the presence of <1% positive cells was classified as a negative result [44]. 

### 2.5. Immunofluorescence for Cell Lines

For Ishikawa and MFE-280 cell lines, incubation with 50 µg/mL of the membrane-staining reagent FITC-conjugated ConA (Merck) was performed for 1 h at room temperature, and cells were washed with ice-cold isotonic PBS [45]. Fixation was performed in acetone and methanol (1:1) for 15 min, followed by PBS washes. Fixed cells were incubated with 1% BSA and 0.1% Triton in PBS (blocking buffer) for 1 h. Primary antibodies anti-hAQP1 (Sigma Aldrich #A5560, 1.6 µg/mL, used at 1:100); anti-hAQP4 (Merck #HPA014784-100UL, 0.5 µg/mL, used at 1:500); anti-hAQP8 (Merck #HPA046259, 2µg/mL, used at 1:100); and anti-hAQP11 (Abcam #ab122821, 1 µg/mL, used at 1:100) were added for 1 h at room temperature. After PBS washes, secondary goat anti-rabbit antibody tagged with Alexa 568 (CF 568, 2 µg/mL, used at 1:1000; Biotium, Fremont, CA USA) was added for 1 h followed by ice-cold isotonic PBS washes. Cells were incubated with Hoescht (Sigma Aldrich # 861405) to stain nuclear DNA for 15 min [46]. Ibidi slides were imaged with an Olympus FV3000 Confocal Microscope (Tokyo, Japan). For Hoechst 33258, the settings were excitation (Ex = 405 nm) and emission (Em = 461 nm). For Alexa Fluor 568, settings were Ex = 561 nm and Em = 603 nm. For ConA, settings were Ex = 488 nm and Em = 513 nm.

### 2.6. Quantitative PCR Analyses for EC Cell Lines

RNA was extracted from untreated (UT), vehicle control (VC; 0.1% ethanol), estradiol treated (1 nM for Ishikawa and 100 pM for MFE-280), and progesterone (100 nM) treated for 48 h, using the RNeasy kit (Cat: 74004; Qiagen, Clayton VIC Australia) as per manufacturer’s instructions. RNA was eluted in 30 µL of RNAse-free water. Quantification of total RNA was performed by Nanodrop (Thermofisher; Waltham MA USA) with integrity confirmed by 260/280 and 260/230 ratios in the range 1.9–2.1, as well as by 18S and 28S bands on a 2% agarose gel, as prerequisites for sample use in downstream applications. RNA (1.5 µg) was reverse transcribed to cDNA using Qiagen Quantitect cDNA conversion kit. Briefly, RNA was heated in gDNA-wipe buffer for 3 min at 42 °C. After adding reverse transcriptase and primers in a final volume of 20 µL, the samples were incubated at 42 °C for 15 min and 95 °C for 2 min.

Quantitative polymerase chain reaction (qPCR) was performed using a Rotor Gene 3000 (Qiagen, Hilden, Germany). Reactions were carried out in a final volume of 10 µL with 5 µL SYBR Green (KapaBiosystems; Wilmington MA USA), 1 µL (1:100 dilution) of cDNA, 1 µL of 10 mM forward and reverse primer mixture, and 3 µL of RNAse-free water. qPCR reactions consisted of initial denaturation at 95 °C for 3 min followed by 40 cycles at 95 °C for 3 s, with annealing and extension at 60 °C for 20 s. Analyses were performed by 2ddCt method [47] relative to the genes *IPO8* (Importin 8) and *PSMC4* (proteasome 26S subunit, ATPase 4) used as reference genes [48,49]. Various reference genes (*18S*, *RPL30*, *TBP*, *GADPH*, *ALAS* and *PUM1*) were also tested; *IPO8* and *PSMC4* were stable during estradiol and progesterone treatments, consistent with published work [48,49]. Sampling was performed in triplicate; independent experiments were repeated twice with independent RNA extractions. Primers for *AQPs 0-12* were designed using NCBI Primer blast (https://www.ncbi.nlm.nih.gov/tools/primer-blast/ (accessed as needed 2019–2023)) and efficiency checked by net primer (https://www.premierbiosoft.com/netprimer/ (accessed as needed 2019–2023)). Selected primers (details shown in Table 2) were obtained from Merck. Prior to qPCR, primers were tested on DNA plasmids using standard PCR to confirm efficacy.

### 2.7. Analyses of Viability in EC Cell Lines

Ishikawa and MFE-280 were cultured in 96-well plates at 10,000 cells/well in 100 µL of culture media for 24 h. Test compounds were added to the wells at final experimental concentrations, and, after 24 h incubation, the MTT (3-(4,5-dimethylthiazol-2-yl)-2,5-diphenyltetrazolium bromide tetrazolium) [50] assay was performed. Briefly, the MTT reagent (Merck) was diluted 1/10 in phenol red-free DMEM; 100 µL of MTT solution was added to each well and plates were incubated 4.5 h at 37 °C in 5% CO_2_. After removing the MTT solution, 100 µL of DMSO was added and plates were wrapped in foil and set on a shaker for 5 min. Plates were read on a BioTek Synergy HTX Microplate Reader at 570 nm. A reference sample with DMSO only (no cells) was included to establish the background signal level.

### 2.8. Transwell Invasion in EC Biopsy Cells and Cell Lines

Transwell invasion tests were carried out using Boyden chambers (6.5 mm Transwell polycarbonate membrane cell culture inserts, 8 μm pore size; Merck #3422). Apical sides of filters were layered with 40 µL of water-diluted (final concentration 0.025 mg/mL) ice-cold extracellular matrix (Matrigel; E1270-10ML) and allowed to set overnight. Matrigel was rehydrated with 50 µL of serum-free medium and incubated 60 min. Ishikawa, MFE-280 and dissociated human EC biopsy cells were trypsinized (0.25% Trypsin/EDTA, Thermofisher Scientific #15400), rinsed, and suspended in serum-free media. Cells (200,000 for cell lines; 100,000 for primary cancer cells) were added to the upper chamber in the presence of the appropriate drug-treatment or vehicle control medium. The lower chamber contained 10% FBS in 700 µL of media, which acted as chemoattractant. Plates were incubated overnight at 37 °C in 5% CO_2_. After 24 h, non-migrated cells were removed gently with cotton swabs from the upper (cis side) surfaces, and invaded cells on the trans side were fixed with 70% ethanol and stained with 0.4% crystal violet (Merck). Invaded cells were imaged with EOS Utility 3 (Canon, Huntington, NY, USA) on an inverted microscope (ULWCD 0.30, Olympus Corp., Tokyo, Japan) at 10× objective [51] and counted. Two independent experiments were performed with 2 to 4 replicates each. Incubation times were 24 h, for Ishikawa and high-grade primary cancer cells, and 30 h, for MFE-280 and low-grade primary cancer cells.

### 2.9. Western Blot

Western blotting was performed with standard protocols [52,53]. Total protein was extracted from Ishikawa and MFE-280 using RIPA (Radio-Immunoprecipitation Assay) lysis buffer with 1% protease inhibitor (Halt) and quantified with the Pierce 660 nm Protein Assay Kit (all reagents from ThermoFisher Scientific). Briefly, 20 µg (for AQP1 and AQP4) or 50 µg (for AQP8 and AQP11) of protein was mixed with Laemmli sample buffer (cat # 1610737, Bio-Rad) and incubated at 65 °C for 10 min. Samples were loaded onto 12% mini-PROTEAN TGX gels (Bio-Rad; South Granville, NSW, Australia) and electrophoresed at 200 V for 25 min. Proteins were transferred to nitrocellulose membrane (GE Healthcare; Parramatta, NSW, Australia) in transfer buffer (25 mM 185 Tris, 192 mM glycine, 20% (*v*/*v*) methanol) for 1 h at 4 °C. Anti-AQP antibodies for Western blot were obtained from Abcam or Merck, as indicated. Primary antibodies (1:1000; using the same antibodies specified above for immunostaining) were added, incubated overnight at 4°, and followed by addition of goat anti-rabbit secondary antibody (Abcam #ab216773, at 1:5000) for 1 h at room temperature. The same membranes were stripped [54] and re-probed with antibody to α-tubulin (Abcam #ab7291, 2 µg/mL), then visualized with a secondary antibody (Abcam #ab216772, 0.2 µg/mL). Membrane images were scanned using the LI-COR Odyssey CLx Imager (LI-COR Biosciences; Lincoln, NE, USA). Protein signal intensities were quantified using Image J [55] and standardized to intensities of the reference protein α-tubulin.

### 2.10. The Kaplan–Meier Survival Analysis

The prognostic significance of *AQP* transcript levels in endometrial carcinomas was analyzed using the Kaplan–Meier (KM) Plotter online database (http://kmplot.com/analysis/ (accessed 2022–2023)) [56]. Associations between progression-free survival and levels of transcripts for selected *AQP* genes were investigated in low- and high-grade endometrial cancers with data from 304 and 1510 cases, respectively. 

### 2.11. Statistical Analyses

Distribution normality was tested by D’Agostino–Pearson tests. One-way ANOVA tested for significant differences across groups compared to vehicle control, followed by post-hoc analyses (unpaired *t*-test for parametric; Mann–Whitney for non-parametric, as stated in Figure legends) using GraphPad Prism 9.0.0. In the box plots, rectangles contain half the values, whiskers show the full range, and horizontal lines indicate median values. Averaged data in x-y plots are mean ± standard deviation. Statistical significance is represented by asterisks as **** *p* < 0.0001, *** *p* < 0.001, ** *p* < 0.01, * *p* < 0.05, and ns (not significant).

## 3. Results

### 3.1. AQP1 Is Highly Expressed in Both Low- and High-Grade EC Tissues

AQP1 expression in human EC tissues was assessed using immunohistochemistry (Figure 1). AQP1 was highly expressed in the EC tissues (Figure 1A). Membrane localization of AQP1 was confirmed in a subset of samples (with patient identification labels Pt.1, Pt.5, and Pt.7). In some cases, variability was evident in AQP1 expression patterns within single cancer samples, suggesting heterogeneity within these tissue samples (Figure 1A; Pt.4, Pt.5 and Pt.7). AQP1 expression, measured as H-scores ranging from 200 to 300 for all EC tissues, indicated high levels of expression of AQP1 (Figure 1B). H-scores were not significantly different between low- (*n* = 5) and high- (*n* = 3) grade EC tissues (Figure 1B). 

The levels of expression of AQPs 4, 8, and 11, assessed in sectioned EC tissues using immunohistochemistry (Figure 2), were weaker than that for AQP1 in both the low- and high-grade cancer tissues (H-scores < 100) (Figure 2A). There were no differences in expression of AQPs 4, 8, and 11 between the tumor grades (Figure 2B). 

### 3.2. High AQP 1, 4 and 11 Transcript Levels Are Related to Poor Prognoses in Grade 3 EC

The Kaplan–Meier online database tool allows assessment of correlations between expression levels of gene transcripts and patient survival [56]. Kaplan–Meier analysis was used to evaluate associations between the *AQP* transcript levels and the overall survival or risk of death (Hazard Ratio, HR) in patients with EC (Table 3). Numbers of cases available for grade 1 (*n* = 304) were too low to provide statistically valid results; however, numbers of cases for EC classified as grade 3 (*n* = 1510) with RNA seq data showed that higher levels of *AQPs 1*, *4*, and *11*, but not *AQP8*, were associated with 1.8- to 2.2-fold increased risk of death.

### 3.3. Plasma Membrane Expression of AQPs 1, 4, and 11 in Ishikawa and MFE-280 Cells

Subcellular localization patterns of AQPs 1, 4, 8, and 11 were assessed by co-staining immunolabeled Ishikawa and MFE-280 cells (Figure 3) with a fluorescence-tagged lectin (FITC-conjugated ConA), which binds glycosylated molecules as a marker for plasma membrane, and Hoechst stain as a marker for cell nuclei. AQPs 1, 4, and 11 were located in the plasma membranes and intracellularly in both cell lines. Using HALO software (module FL V 2.1.3, IndicaLab, Albuquerque, NM, USA), colocalization of AQP channels and ConA (yellow signals) was observed for AQPs 1, 4, and 11 in Ishikawa and MFE-280 cells (Figure 3A,B). AQP8, faint (despite a high anti-AQP8 antibody concentration, 1:100 dilution) but present in Ishikawa cells, was intracellular without detectable plasma membrane colocalization. For visualization only, all eight AQP images (3rd column, ‘AQP’) were standardized in unison using GIMP 2.8 image software (Charlotte, NC, USA), using the threshold tool to expand grayscale values to the full Gaussian range for all images combined (result shown as ‘AQP thresh’ in the 4th column); this analysis clearly shows that AQP8 is present in Ishikawa but not evident in MFE-280 cells. 

### 3.4. Comparison of Pharmacological Modulator Effects in Reducing Invasiveness of EC Cell Lines

An array of pharmacological agents previously suggested to inhibit AQP channel activities were tested for the effects on invasiveness of Ishikawa and MFE-280 cells using transwell invasion assay (Figure 4). Numbers of cells that penetrated through an extracellular matrix-layered filter to reach the trans-side of filter chambers are summarized in box plots, with data standardized to invasiveness in vehicle control (0.1% DMSO; set as 100%), which was not different from untreated controls.

In both Ishikawa (Figure 4A) and MFE-280 (Figure 4B) cell lines, the most potent blockers of transwell invasion were 5-PMFC, curcumin, and resveratrol. 5-PMFC is known to block the AQP1 cation conductance through the central pore, but does not impair osmotic water flux through the parallel intrasubunit pores [29]. Transwell invasion of Ishikawa cells was blocked by AQP1 ion channel inhibitors, being reduced ~41% with 80 µM AqB011 and ~55% with 0.5 mM 5-PMFC. In MFE-280 cells, AqB011 at 60 µM (the maximum dose tolerated in this cell line) showed a trend towards reduced invasion (~18%; not significant), and 5-PMFC at 0.5 mM inhibited invasion by ~77%. These results suggest both cell lines rely on AQP1-mediated ion channel activity as part of the invasion process. 

In contrast, agents that have been proposed to decrease AQP water channel activity (acetazolamide, ginsenoside Rg3, Keen Mind, TGN-020, and IMD-0354) did not impair invasiveness in either Ishikawa or MFE-280 cell lines. Unexpectedly, ginsenoside Rg3 and IMD-0354 boosted transwell motility uniquely in MFE-280 cells, which might reflect an accelerated metabolic activity measured in MTT assays (below). 

The broad-spectrum agents curcumin and resveratrol inhibited invasion by ~30 and ~71% in Ishikawa, and by ~65 and ~71% in MFE-280. Diverse ion channels and transporters have been implicated in responses to curcumin (AQP4, K^+^, Ca^2+^, anion channels, glucose transporters [58]) and resveratrol (including AQP3 [58]) and other signaling molecules. Some of the apparent inhibitory effects of curcumin on invasion might reflect cytotoxicity in Ishikawa cells. Representative images show the effects of treatments on the invasiveness of cell lines (Figure 4C,D).

In a spheroid preparation for assessing invasiveness, AqB011 significantly reduced the rate of invasion of Ishikawa cells into the surrounding matrigel (Appendix A).

### 3.5. AQP1 Ion Channel Inhibitors Restrained Invasiveness of Primary EC Cells

The two AQP1 ion channel blockers, 5-PMFC and AqB011, which were effective in reducing invasion of EC cell lines, were tested on primary EC cells derived from patient tissues (Figure 5). Numbers of cells penetrating through an extracellular matrix layer were compiled as box plots (Figure 5A,B). 5-PMFC (0.5 mM) caused 40 to 50% inhibition of invasion in primary cultures of high- and low-grade cancer cells. AqB011 (80 µM) showed more variability; levels of inhibition of invasion ranged from 15% to 50% in high- and low-grade cancer cells. Images representing the invasion of control and treated groups are shown in Figure 5C,D. AQP1 ion channel inhibitors reduced invasiveness of human primary EC cancers established from both low- and high-grade tumors.

Variability in the effectiveness of AqB011 between tumor tissues was investigated by comparing levels of AQP1 protein expression and the inhibitory effect of AqB011 within each sample. A robust positive linear correlation showed that increased AqB011 effectiveness corresponded to the increased level of AQP1 expression in primary EC cells (Figure 5E). 

### 3.6. Aquaporin Expression (RNA and Protein) in Ishikawa and MFE-280 Cells Is Regulated by Estradiol and Progesterone

RT-qPCR was used to quantify transcript levels for all classes of human aquaporins in both cell lines. The histogram (Figure 6) shows mean aquaporin transcript levels in Ishikawa and MFE-280 cell lines, calculated relative to stable reference genes *IPO8* and *PSMC4* [48,49], which were set as 1.0. In the non-stimulated state, Ishikawa cells showed higher levels of transcripts for *AQPs 0*, *4, and 8* as compared with MFE-280; conversely, MFE-280 cells showed higher levels of transcripts for *AQPs 1*, *2*, *3*, *5*, *11*, and *12B* as compared with Ishikawa. *AQPs 6*, *7*, *9,* and *10* were not detected (ND) in either cell line, as defined by Ct threshold values > 33, multiple peaks in the melt curves, and high variability (>1 cycle) in Ct values between triplicates. Primer efficacies were confirmed in advance by standard PCR, which showed successful amplification of correct bands from *AQP0-12* plasmid DNAs.

Previous work showed that estradiol treatments potentiated the transwell invasiveness of Ishikawa cells at an optimal concentration of 1 nM and MFE-280 at an optimal concentration of 100 pM, whereas invasion was inhibited in Ishikawa cells by 100 nM progesterone, which had no effect on MFE-280 [59]. A possible role for AQPs as mediators of the pro- and anti-invasive responses to hormone signaling has not previously been evaluated. Assays here tested the effects of estradiol and progesterone on transcript levels of *AQPs* measured in Ishikawa and MFE-280 cell lines, comparing responses to treatments with estradiol (100 pM, 1 nM) and progesterone (100 nM) as compared with vehicle control at 48 h (Figure 7). Data were standardized to reference genes, as per protocol, and then normalized to vehicle control. Both cell lines were incubated for 48 h with physiological doses of estradiol (100 pM, 1 nM) and progesterone (100 nM) and tested for changes in gene expression of aquaporins 0 to 12. Ishikawa cells treated with estradiol showed upregulation of *AQPs 5*, *11*, and *12*, and downregulation of *AQPs 0* and *4*. MFE-280 cells showed a different response to estradiol with increased levels of *AQPs 0*, *1*, *3*, and *8*, and decreased *AQP11*.

A subset of AQP classes was selected for in-depth analyses based on *AQP* transcript patterns, characterized in Figure 1 and Figure 2, above, and prognostic significance as summarized above in Table 3. AQP1 channels have been well-documented in published studies to be linked to mechanisms of cell motility in other cancer types, and they have successfully been targeted by pharmacological agents that dramatically slowed cancer migration and invasiveness, as reviewed previously [10,11]. AQP1 has not previously been assessed for a role in EC cell motility, and, hence, was selected for inclusion here as benchmark for comparison. In addition, a subset of three additional aquaporins not previously identified in human endometrial cancers were chosen based on their distinctive patterns of regulation by hormone signaling, a key factor in EC progression. These were: AQP4, which, in response to estradiol, was increased in MFE-280 and decreased in Ishikawa; AQP8, which showed unusually high expression levels and was decreased by progesterone in Ishikawa and increased by estradiol in MFE-280; and AQP11, which was increased by estradiol in Ishikawa and decreased in MFE-280.

The levels of protein expression for AQPs 1, 4, 8, and 11 were assessed for Ishikawa and MFE-280 by Western blot (Figure 8), quantified with reference to the housekeeping protein, α-tubulin. The Western blot analysis confirmed that all four selected classes of AQPs were expressed at the protein level in both cell lines. Estradiol and progesterone mediated upregulation of AQP1 in Ishikawa cells (Figure 8A,B). Otherwise, the overall protein levels remained relatively constant across the treatment paradigms (Figure 8C–H). The high molecular weight bands for AQP4 (75 kDa detected vs 35–45 kDa expected) and AQP8 (100 kDa detected vs. 28 kDa expected) have been reported previously and suggested to reflect glycosylation, dimerization of subunits, or both [60,61,62]. Full Western membrane views are provided in Appendix A. Apparently, higher protein levels detected by Western blot for Ishikawa as compared with immunocytochemistry might suggest that other factors influence the availability of the anti-AQP8 epitope site in the immunocytochemistry preparation.

### 3.7. Effects of Pharmacological Treatments on EC Cell Viability

Cell viability was measured as mitochondrial metabolic activity using the MTT assay (Figure 9). The pharmacological agent 5-PMFC at 24 h did not impair viability of Ishikawa or MFE-280 cells at the dose used to control invasiveness (0.5 mM), but it did reduce viability at 1 mM (Figure 9A). The agent AqB011 at 24 h in Ishikawa showed no toxicity; however, more sensitive MFE-280 cells showed decreased viability at 60 and 80 µM (Figure 9B). Effects of other aquaporin modulators on viability are summarized in Appendix A.

## 4. Discussion

The important outcome of this work was the demonstration that pharmacological blockade of AQP1 ion channels substantially inhibited invasion both in EC cell lines and in primary EC cells derived from patient low- and high-grade tumors. The AQP1 ion channel blockers used in this study were AqB011 [11,20,28,64], previously shown to inhibit migration and invasion in AQP1-expressing colon cancers [11,12], and the furan compounds 5-HMF and 5-PMFC, shown to block the AQP1 cation conductance in red blood cells at 0.5 to 1 mM, as well as to reduce colon cancer motility at similar concentrations [29,30]. AqB011 significantly restrained the invasion of primary EC cells in biopsy samples of low- and high-grade ECs. AqB011 is specific for inhibiting the ion channel activity of AQP1 with no effect on parallel water channel permeation [11,28]. Concentrations of these compounds that blocked EC cell invasion aligned with the concentrations shown to block the human AQP1 ion conductance in electrophysiology assays [11,29]. The more subtle effect of AqB011 on MFE-280 invasiveness (~20% block at 60 µM) as compared to Ishikawa might reflect the lower dose used, since 80 µM, unusually, was toxic for MFE-280. In addition, the comparatively high levels of AQP1 protein in MFE-280 could make it difficult for AqB011 to block a rate-limiting proportion of the channel population. 5-PMFC showed a relatively consistent degree of inhibition (~50%). The inhibitory effects of 5-PMFC might be potentiated by parallel actions on other targets; this agent is not selective for AQP1 only. 5-PMFC was shown to exert beneficial effects by a double action in blocking the AQP1 cation current while inducing structural modifications in other key proteins [29].

Plasma membrane localization of AQP1 in the Ishikawa and MFE-280 (confirmed by immunolabelling) is consistent with a role for AQP1 ion channel signaling and invasiveness. Pharmacological blockers of AQP1 ion channel activity impaired transwell invasion of EC cells. AQP1, polarized to leading edges of migrating cells, is thought to enable lamellipodial protrusion and cell movement [6,65]. Only a small proportion of active AQP1 ion channels in a cell membrane was calculated to be sufficient to drive physiologically relevant changes in membrane potential and to increase net Na^+^ re-absorption in the kidney proximal tubule [66]. Interaction of cyclic GMP (involved in tumorigenesis and metastasis of cancers such as colon [67], breast [68], and cervical [69]) with the intracellular gating loop of AQP1 opens the AQP1 central pore and serves as the site of action by the inhibitor AqB011 [28,70].

Variation in AQP1 protein expression was observed in tissues from patients for both low- and high-grade ECs. A heterogeneous pattern of AQP1 expression within the same tissues was also observed, indicating variability in the levels and distributions of AQP1 protein within single tumor masses. Intra-tumoral heterogeneity is well-known in EC, in which variable protein expression patterns can be observed within the same patient samples [71,72]. The basis for the molecular heterogeneity within the same tumor is not known, but could reflect genomic instability [73] and contribute to resistance to therapy [72]. We found that patient tissues with non-uniform distributions of AQP1 protein also showed greater variability in primary EC cell invasiveness for replicates within the same experiment and between duplicate experiments. The observed inter- and intra-tumoral heterogeneity and interpatient variability of AQP1 expression emphasizes the importance of combining multiple approaches in cancer treatment. Discrepancies between transcript and protein levels for a given gene product have been described previously [74,75]. The levels of AQP protein expression in EC cells reflect a dynamic balance between processes of transcription, translation, and protein turnover. AQP posttranslational modification and localization ultimately define their functional roles in endometrial homeostasis and EC pathophysiology.

5-PMFC is a more potent inhibitor of the human AQP1 ion channel than 5-HMF, as shown by voltage clamp electrophysiology using the *Xenopus* oocyte expression system [30]. 5-PMFC caused a significant reduction of invasion in the EC cell lines at 0.5 mM, whereas 5-HMF required a dose of 1 mM to achieve anti-invasive activity in Ishikawa; these results match the dose-dependent effects of these agents on the AQP1 ion conductance and anti-sickling activities in red blood cells [29]. Some compounds tested here that did not demonstrate appreciable effects on EC invasiveness have been found to exert anti-cancer activities in other models, suggesting selective treatments with AQP antagonists might need to be matched to cancer subtypes. For example, ginsenoside reduced motility in breast and prostate cancer cells [8,10]; bacopasides reduced colon cancer cell motility [12]; and acetazolamide reduced angiogenesis [76].

Traditional Chinese Medicines (TCMs), known for multifaceted effects on diverse cellular targets (including aquaporin modulation), also showed promising effects, particularly for resveratrol and curcumin, which exerted robust anti-invasive control in both EC cell lines. Confirming their long-standing traditional popularity in alternative medicines, TCMs have anti-cancer benefits in a variety of cancer subtypes [33,34,77,78,79,80]. In EC cells, resveratrol was effective at non-toxic concentrations. Conversely, curcumin appeared to cause significant inhibition of invasion in both EC cell lines but was toxic in Ishikawa cells, raising the caveat that some of the apparent reduction in invasion was an indirect consequence of cell death. Determining whether resveratrol, among its many roles, also blocks AQP1 ion channels will be an interesting area for further work.

Physiologically relevant doses of estradiol and progesterone were found to differentially regulate the levels of expression of different classes of aquaporins in low- and high-grade EC cells, which could contribute to some of the observed heterogeneity in expression and pharmacological sensitivity between EC samples. Estradiol, known to be pivotal to the progression and spread of EC [81], was reported previously to increase AQP1 expression in Ishikawa cells. Estrogen response elements (EREs) and progesterone response elements (PREs) have been identified in the DNA promotor regions of genes encoding *AQPs 1*, *2*, and *3* [17,82,83]. Estradiol upregulates *AQP1* in prostate cancer [8]; *AQP2* in EC [17]; and *AQP3* in breast cancer [82]. Similarly, the expression of *AQP1* was enhanced in MFE-280 cells by estradiol but not progesterone, in agreement with prior work showing that estrogen receptor (*ESR1* and *ESR2*) transcripts are present in the MFE-280 cell line, but these cells do not express detectable levels of progesterone receptor isoforms (PRA or PRB) as assessed by Western blot [59].

In contrast to transwell results, MFE-280 cell lines showed little invasive behavior as measured by the dissemination of particles away from spheroids (Appendix A), perhaps reflecting additional factors that impact dispersal from an aggregated spheroid, such as cell adhesion and cell–cell interactions [51,84]. In addition, the static extracellular matrix used for spheroids lacks a chemoattractant gradient comparable to that used in the transwell method. Atypical behavior in the spheroid model has precedent from work with the breast cancer line MCF-7, which, though highly invasive, did not exhibit dispersal in the spheroid system [85]. This assay remains to be optimized as a tool for monitoring in vivo cancer behavior.

The association between AQP expression levels and patient outcomes for high-grade EC was explored by analyzing RNAseq data from the KM online database, which indicated that high levels of *AQPs 1*, *4*, and *11* transcripts were associated with reduced patient overall survival. *AQP1* was shown to be a prognostic marker for poor outcomes in epithelial ovarian cancer and in highly invasive basal-like breast cancer [86,87]. Similarities in genetic and histological features between endometrial, epithelial ovarian, and breast cancers [40] suggest *AQP1* might have prognostic significance across multiple classes of hormone-sensitive cancers. An increase in *AQP1* expression levels with increasing cancer grade, coupled with plasma membrane localization, upregulation by estradiol, and inhibition of invasiveness by AQP1 ion channel blockers, converge on the interesting proposal that AQP1 ion channel function is important for EC progression. Pharmacological blocking of AQP1 channels as a means to control disease processes has attracted interest [11,29], but the selectivity and potency of effective pharmacological agents remain to be refined and optimized.

AQPs 1, 4, 8, and 11 were chosen on basis of their novel identification in EC, associations with poor prognoses, and distinctive patterns of regulation by hormone signaling. Membrane localization of AQP1 was observed in both EC cell lines. Although high *AQP 4* and *11* gene expression has been linked to poor prognosis in grade 3 ECs (Table 3), pharmacological assays with proposed AQP4 inhibitors, such as TGN-020, did not constrain invasive behavior in either of the EC cell lines (Figure 4). On the contrary, IMD-0354 boosted the invasiveness of MFE-280 cells. Protein expression of AQP8 was significantly decreased in high-grade EC cell lines. Overexpression of AQP8 has been linked with reduced migration and invasion through inhibition of PI3/Akt in colon cancer [88], and pathologies such as epithelial ovarian, cervical, colorectal, and liver cancers also have been associated with altered AQP8 expression [5,89,90,91]. Clinical trials indicated high expression of AQP8 was associated with better prognoses and overall survival in colon cancer [92]. *AQP11* transcripts, at high levels, have been associated with poor survival in lung adenocarcinoma [93] but, conversely, better survival in breast cancer [94], suggesting cancer-specific contributions of different classes of aquaporins. Selective inhibitors for AQPs 8 and 11 remain to be identified; in the interim, interfering RNA knockdown techniques might be used to explore the possible roles of these channels in EC cells.

MFE-280 is representative of a high-grade metastatic EC that is not responsive to progestin treatment. Prior work confirmed that MFE-280 cells do not express progesterone receptors; in agreement, invasiveness in this cell line was not inhibited by progesterone (1 pM to 100 nM) [59]. The potential opportunity to limit invasiveness of progestin-insensitive cells using an ion channel blocker of AQP1 merits exploration using chemical structure–activity analyses to develop analogs with higher potency and low toxicity. Cross talk between tumor and stromal cells is known to enhance EC invasiveness and growth [95]; this limitation was addressed in part here by using human biopsy samples, which provided more complex microenvironments. Primary EC cells derived from patient biopsies are closer to an accurate representation of EC tumor cells, retaining stem-like phenotypes, and their protein and transporter expressions are similar to the in vivo state [96,97].The validation here, that AQP1 ion channel blockers inhibited invasiveness in primary cancer cells from human biopsies as well as the EC cell lines, suggests both preparations have value for drug discovery efforts. Results here showed the invasiveness of EC cells relies on AQP1 ion channels, and pharmacological inhibition of these channels could be a new strategy to limit motility in resistant cancer types. However, in vivo studies are needed in the future to validate in vitro results of AQP1 ion channel inhibitors.

While the utilization of in vitro studies using cell lines and primary cells offers advantages like accessibility, ease of cultivation, and cost-effectiveness for drug testing, they lack the key components involved in cancer survival and metastasis, such as stromal cells, immune cells, and other cytokines and growth factors [98,99]. Future directions should be focused on validating the results in vivo and expanding the patient numbers to enhance the robustness of our findings. The dose of AqB011 and 5-PMFC used in this research had been successfully used in rodent in vivo models. Previously, AqB011-treated mice were used to assess the role of AQP1 in cardiac hypertrophy [100], and 5-PMFC was used at doses up to 2.5 mM in human red blood cells and rodent models of sickle cell disease for anti-sickling effects [29]. Further work on chemical structure–activity relationships, distribution, and metabolism will be needed, along with in vivo studies, for progress towards clinical translation.

## 5. Conclusions

This is the first study to identify the effects of AQP1 ion channel blockers in reducing the invasiveness of EC cells. Combining ion channel inhibitors of AQP1 with hormonal therapy in patients with endometrial cancer could address resistance to hormonal treatments, and could also help reduce the intensity of regular cancer treatments needed, sparing patients from the associated side effects. Based on preliminary work conducted here, using primary and endometrial cancer cell lines in vitro, AqB011 and 5-PMFC are the lead candidate compounds that could be used for the development of new agents to restrain the invasiveness of EC cells, but they will require in vivo testing to fully explore their translational potential.

## Figures and Tables

**Figure 1 cancers-15-04507-f001:**
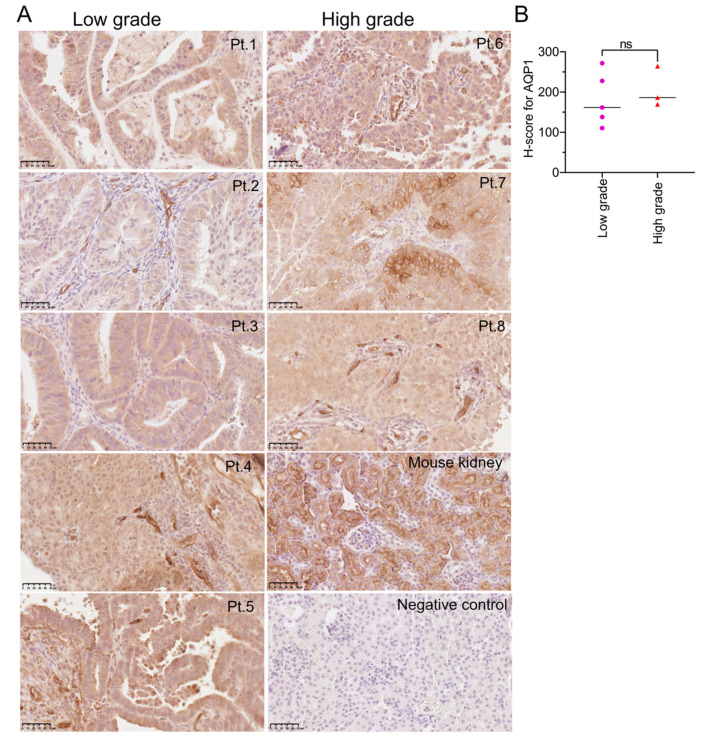
Immunostaining for AQP1 protein in EC tissues. (**A**) AQP1 was detected by immunohistochemistry using horseradish peroxidase (brown staining). The endometrial glands are distinctive and abundant in low-grade cancers (deidentified patient samples labeled Pt.1 to Pt.5). The endometrial cancer cells are present in sheets in high-grade cancer tissues (Pt.6, Pt.7, Pt.8). Membrane localization of AQP1 is evident in the image for Pt.5. High expression of AQP1 was evident in blood vessels (Pt.4 and Pt.8). Scale bars are 50 µm. (**B**) AQP1 H-scores, segregated for low- (*n* = 5) and high- (*n* = 3) grade tissues. AQP1 protein expression was assessed using QuPath software. No significant difference in H-scores was found between low- and high-grade EC tissues (Mann–Whitney test).

**Figure 2 cancers-15-04507-f002:**
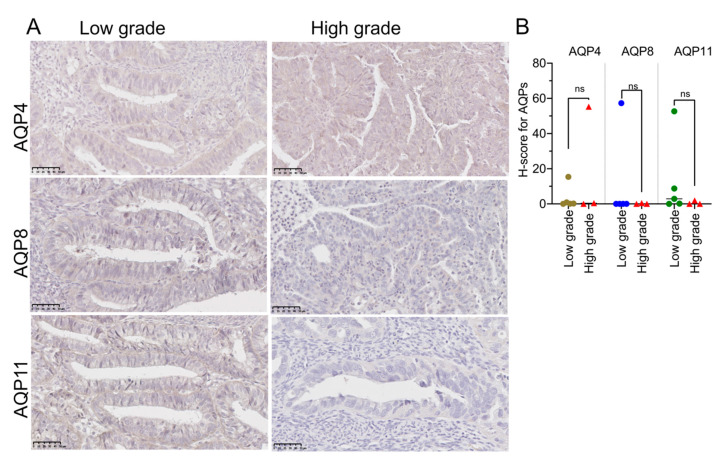
Immunostaining for AQPs 4, 8, 11 protein in EC tissues. (**A**) Representative images of IHC staining for AQPs 4, 8, and 11 detected by immunohistochemistry. Scale bars are 50 µm. (**B**) Quantification of protein expression for low- (*n* = 5) and high- (*n* = 3) grade tissues using QuPath software. No significant difference (Mann-Whitney test) in H-scores was observed between low- and high-grade EC samples for AQPs 4 and 11.

**Figure 3 cancers-15-04507-f003:**
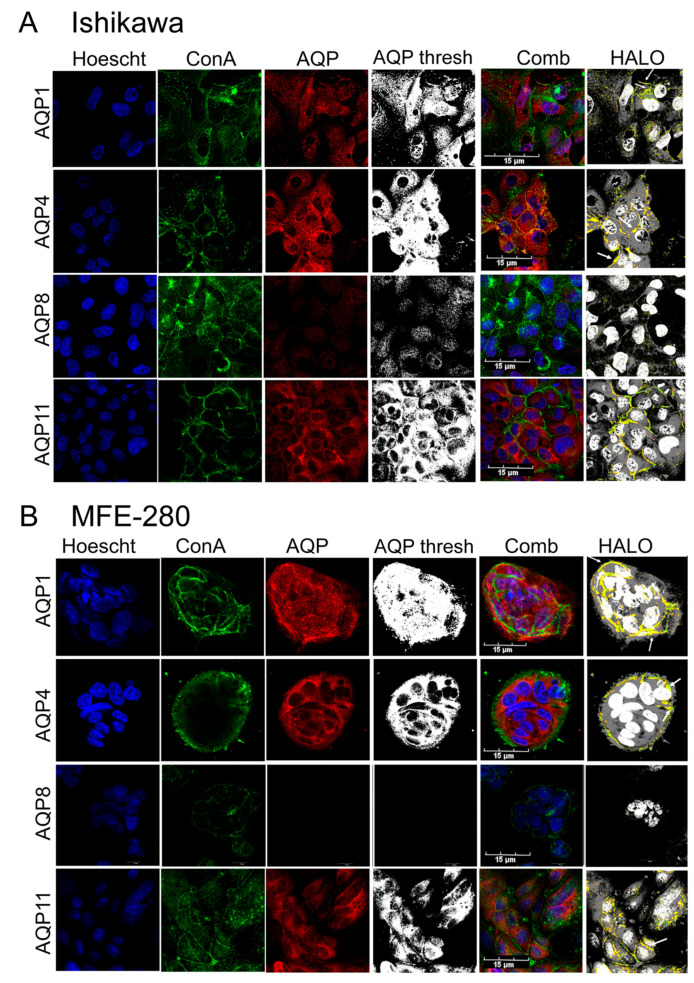
Immunocytochemical localization of AQPs 1, 4, and 11, but not AQP8, in Ishikawa (**A**) and MFE-280 cells (**B**). Nuclei were labeled with Hoescht stain (blue); plasma membranes were marked by a fluorescence-tagged lectin ConA (green) [57]; AQP signals were visualized by primary antibody with fluorescence-tagged secondary antibody (red); and the three signals were superimposed in a combined view (Comb). The ‘AQP thresh’ shows AQP images batch-processed in unison using GIMP software to utilize the full range of pixel intensities from min to max. HALO image analysis identified colocalization (yellow) of AQPs 1, 4, and 11 with the plasma membrane marker ConA in both cell lines. White arrows illustrate regions of strong colocalization. AQP8 was detected at low levels only intracellularly in Ishikawa, and not detectable in MFE-280 cells. Scale bar = 15 µm.

**Figure 4 cancers-15-04507-f004:**
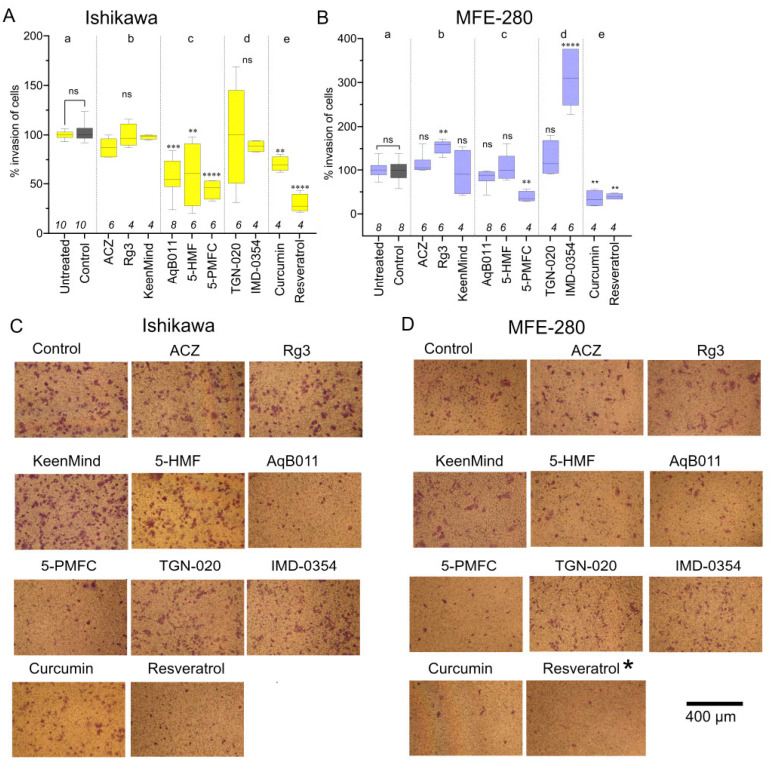
The effects of pharmacological agents (AQP channel inhibitors) on EC cell line invasion. Box plot summaries showing percent invasion of (**A**) Ishikawa after 24 h and (**B**) MFE-280 cells after 30 h treatments in (a) controls, or (b) with proposed AQP water channel inhibitors acetazolamide (ACZ, 1 µM for MFE-280, 10 µM for Ishikawa), ginsenoside Rg3 (100 µM), and KeenMind (44 µM); (c) AQP1 ion channel inhibitors 5-HMF (1.5 mM), AqB011 (60 µM for Ishikawa, 80 µM for MFE-280), 5-PMFC (0.5 mM); (d) proposed AQP water channel inhibitors TGN-020 (3 µM), IMD-0354 (0.2 µM); or (e) broad-spectrum inhibitors curcumin (20 µM) and resveratrol (40 µM). Statistical comparisons used one-way ANOVA and post hoc parametric *t*-tests. ** *p* < 0.01; **** *p* < 0.0001; *** *p* < 0.001; ns, not significant as compared with control. (**C**,**D**) Representative images of invaded cells for (**C**) Ishikawa (24 h) and (**D**) MFE-280 (30 h) with treatments as indicated (same doses as in (**A**,**B**)). Scale bar 400 µm. Asterisk (*) indicates the revised image for Resveratrol.

**Figure 5 cancers-15-04507-f005:**
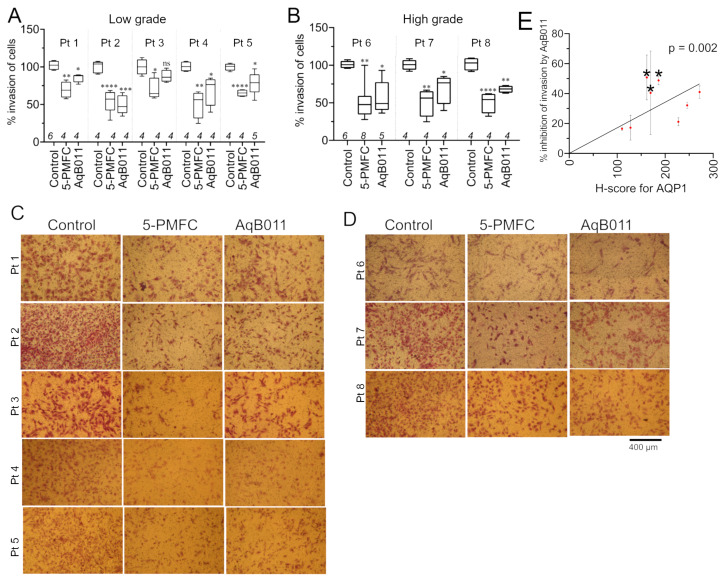
Effects of aquaporin-1 channel blockers on invasiveness of low- and high-grade primary human EC cells. Primary EC cells were treated with 0.5 mM PMFC or 80 µM AqB011 (24 h for low-grade; 18 h for high-grade) and compared with vehicle-treated controls. Box plots show percent invasion for (**A**) low-grade and (**B**) high-grade. Statistical analyses used one way ANOVA and post hoc *t*-tests (* *p* < 0.05; ** *p* < 0.01; *** *p* < 0.001; **** *p* < 0.0001; ns not significant) as compared with control. Representative images of transwell-migrated cells in (**C**) low-grade and (**D**) high-grade patient biopsies for treatments, as labeled. “Pt. *n*” is patient ID number. Scale bar = 400 µm. (**E**) Linear correlation of AQP1 protein expression in human biopsy samples assessed by immunohistochemistry, quantified as H values and plotted against the percent inhibition of transwell invasion by the AQP1 ion channel blocker AqB011 (80 µM; 24 h for low-grade; 18 h for high-grade). Regression analyses performed by GraphPad Prism showed a non-zero slope (*p* < 0.002). Data are mean ± SD.

**Figure 6 cancers-15-04507-f006:**
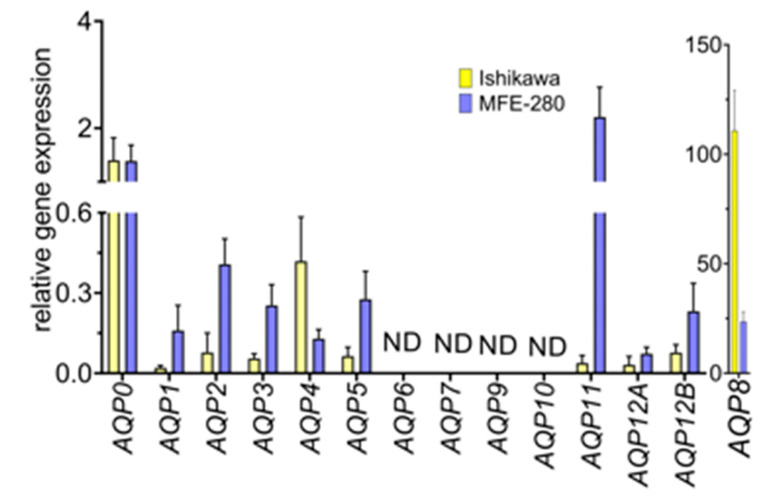
Transcript levels for aquaporins relative to reference genes (I*PO8*, *PSMC4*) in Ishikawa and MFE-280 cells analyzed by qPCR using the Livak (2ddct) method. Histogram data showed multiple classes of *AQP*s are expressed in Ishikawa and MFE-280 cells, with the exception of *AQP*s 6, 7, 9, and 10, which were not detected (ND) in either cell line.

**Figure 7 cancers-15-04507-f007:**
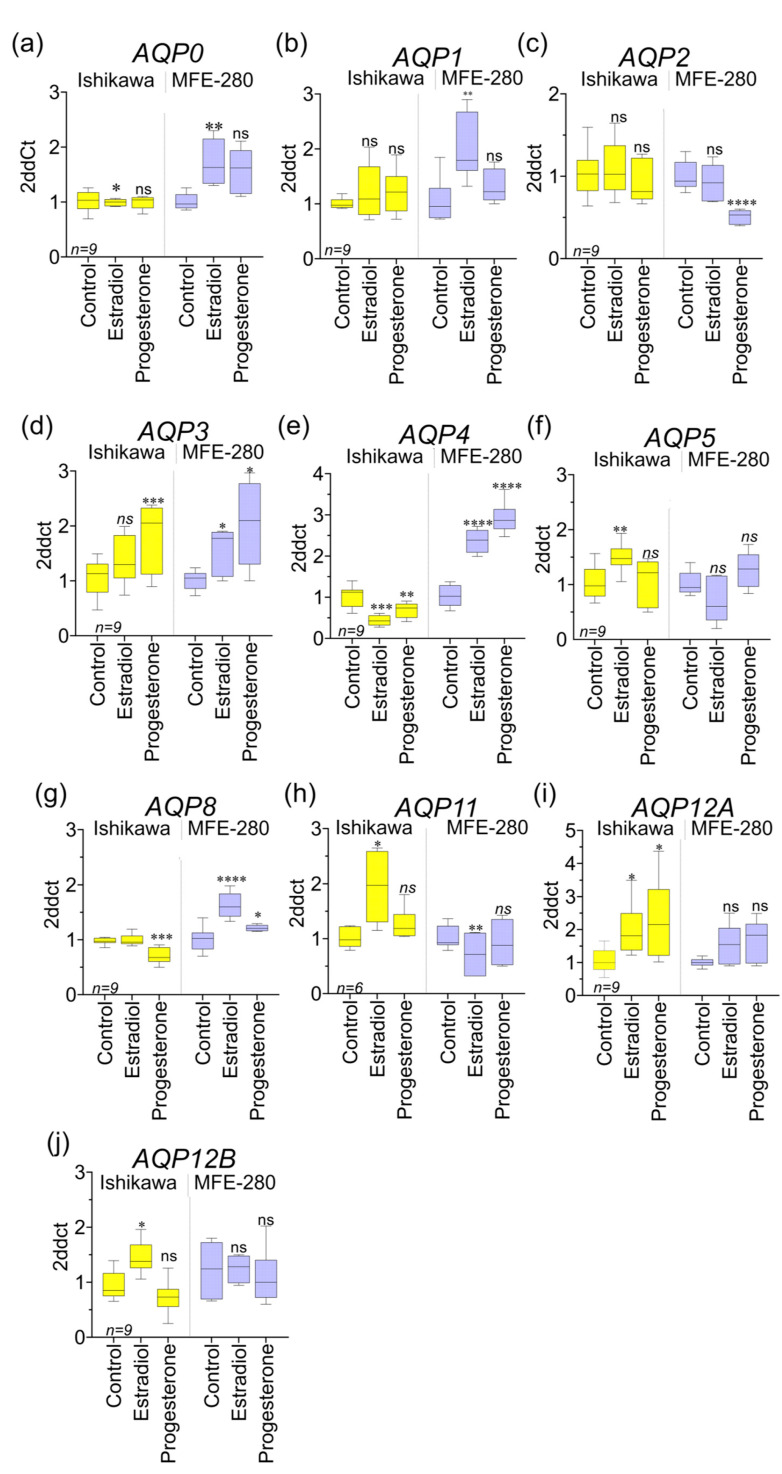
Relative transcript levels for aquaporins in EC in response to estradiol and progesterone. Data summarized in box plots show responses to hormone treatments standardized to matched vehicle control groups for the ten *AQP* subtypes found to be present in Ishikawa and MFE-280 cells. Treatments (48 h) were 100 pM estradiol, 100 nM progesterone, or vehicle control (0.01% ethanol). The Livak (2ddct) method was used for qPCR analyses. A total of 3 independent experiments with 3 replicates each were performed with isolated RNA from different cell line passage numbers. Statistical analyses with one way ANOVA and post hoc *t*-tests (* *p* < 0.05; ** *p* < 0.01; **** *p* < 0.0001; *** *p* < 0.001; ns not significant) as compared with control Box plots (**a**–**j**) show levels of *AQP* transcripts relative to matched vehicle controls for Ishikawa and MFE-280. Both cell lines were shown previously to express estrogen receptors *ESR1* and *ESR2* at the mRNA level. Ishikawa cells also were found to express progesterone receptors PRA and PRB; in contrast, no progesterone receptors were detectable in the MFE-280 line for protein or mRNA [59]. Estradiol treatment resulted in increased transcript levels for *AQPs 5*, *11*, *12A,* and *12B* in Ishikawa cells. A different pattern in MFE-280 showed increased levels of *AQPs 0*, *1*, *3*, *4,* and *8* secondary to estradiol treatment. In terms of downregulation of *AQP* expression, estradiol resulted in decreased levels of transcripts only for *AQP0* and *AQP5* in Ishikawa and only *AQP11* in MFE-280 cells. Progesterone in Ishikawa cells caused upregulation of *AQP3* and *AQP12a* and downregulation of *AQP4* and *AQP8*. In MFE-280, *AQP2* was decreased while *AQPs 3*, *4*, and *8* were increased by progesterone.

**Figure 8 cancers-15-04507-f008:**
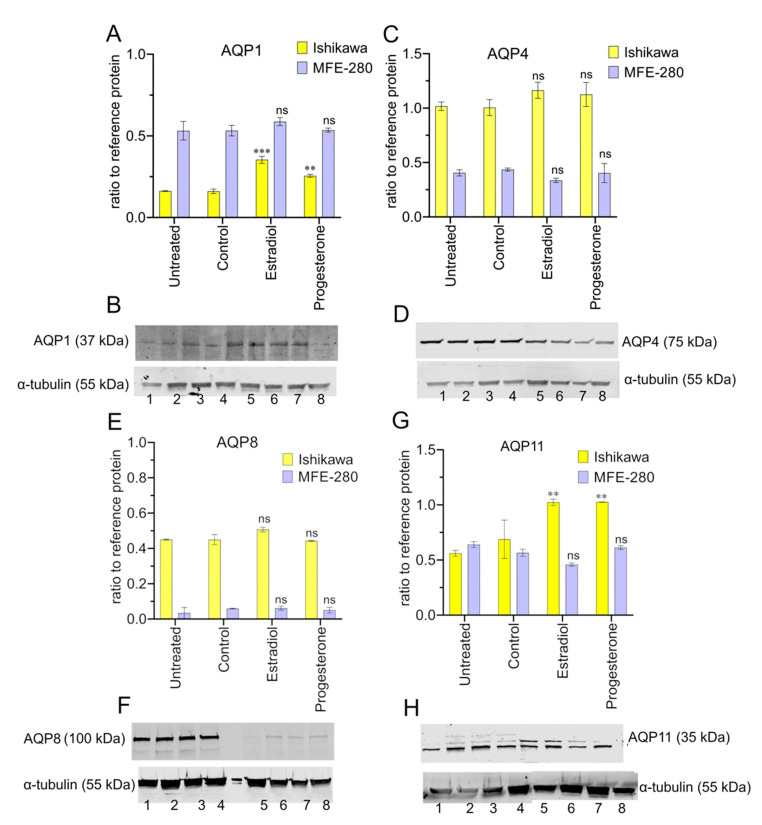
Western blot showing protein expression levels for AQPs 1, 4, 8, and 11 in hormone-treated and control Ishikawa and MFE-280 cells (48 h) as ratio to reference protein, α-tubulin. A total of 20 µg of protein was loaded for AQP1 (**B**) and AQP4 (**D**); 50 µg protein was loaded for AQP8 (**F**) and AQP11 (**H**). Histograms show compiled data for mean AQP signals and corresponding gel images for: AQP1 (**A**,**B**); AQP4 (**C**,**D**); AQP8 (**E**,**F**); and AQP11 (**G**,**H**). Lane numbers represent: 1 = Ishikawa untreated, 2 = Ishikawa control, 3 = Ishikawa 1 nM estradiol, 4 = Ishikawa 100 nM progesterone, 5 = MFE-280 untreated, 6 = MFE-280 control, 7 = MFE-280 100 pM estradiol, 8 = MFE-280 100 nM progesterone. Data are from *n* = 2 independent experiments performed in duplicate with protein samples from different passage numbers. For standardization, membranes were stripped and re-probed for loading protein, α-tubulin. Signal intensities were quantified with Image J 1.54 F software (https://imagej.nih.gov/ij/download.html) [63]. Statistical analyses used one-way ANOVA and post-hoc parametric *t*-tests (** *p* < 0.01; *** *p* < 0.001; ns not significant).

**Figure 9 cancers-15-04507-f009:**
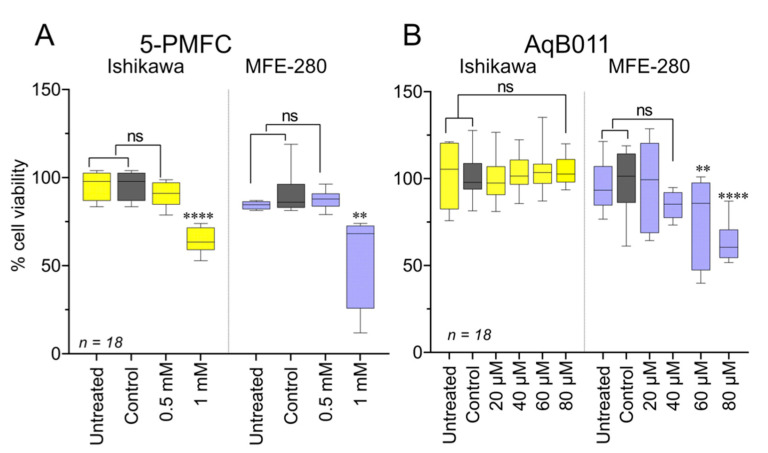
Ishikawa and MFE-280 cell viability was assessed by MTT assay after treatment with AQP1 ion blockers. Box plots for Ishikawa and MFE-280 cells after 24 h treatment with: (**A**) 5-PMFC (0.5–1 mM), (**B**) AqB011 (20–80 µM), each compared with matched controls. Three independent experiments were performed with six replicates each (*n* = 18). Statistical analyses used one way ANOVA with post hoc parametric *t*-tests (** *p* < 0.01; **** *p* < 0.0001; ns not significant).

**Table 1 cancers-15-04507-t001:** Summary of pharmacological agents used to assess possible contributions of aquaporin channels in EC, with structures and experimental doses used.

Compound	Structure	Dose
Acetazolamide	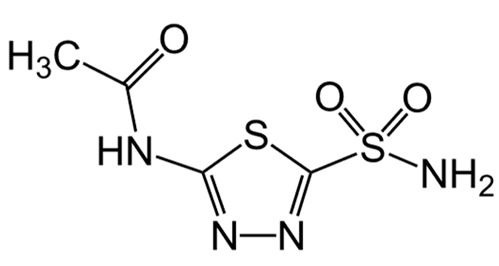	1–100 µM [25,26]
Ginsenoside (Rg3)		100 µM [8,10]
Bacopaside II (KeenMind)		44 µM [27]
AqB011	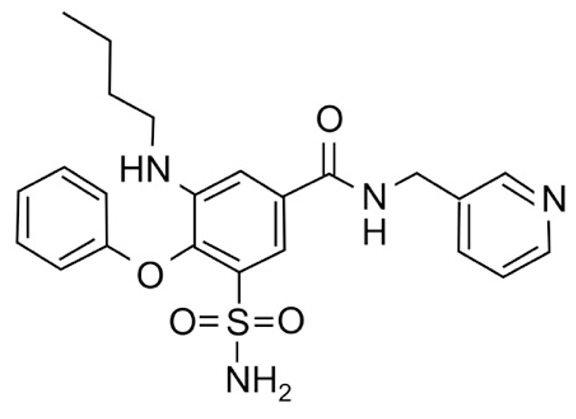	80 µM [11,28]
5-HMF		1 mM [29,30]
5-PMFC	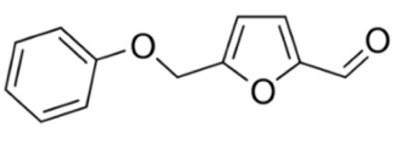	0.5 mM [29,30]
IMD-0354	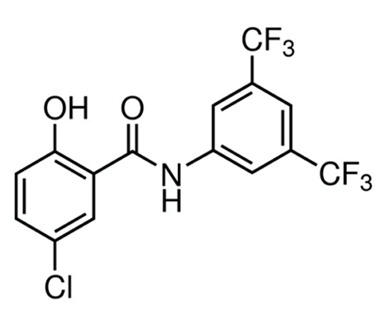	0.2 µM [13]
TGN-020	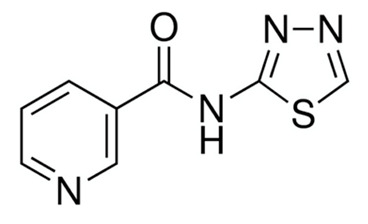	3 µM [32]
Curcumin	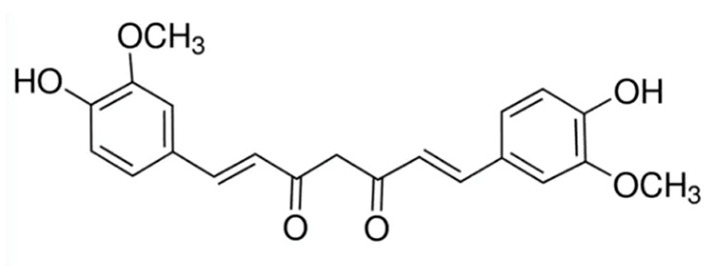	20 µM [33,34]
Resveratrol	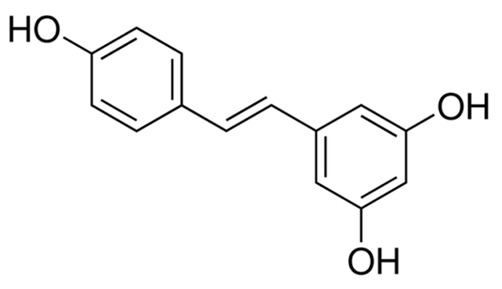	40 µM [35]

**Table 2 cancers-15-04507-t002:** Primer sequences used for aquaporin subtype-selective transcript amplification.

Gene	Primer Sequence	Product Size *	T_M_ °C	Gene ID Number
*AQP0*	F = CTAGCACTCAACACGTTGCAC	210	60.1	NM_012064.4
	R = AGGATTCATGCCTGCACCAG		60.4	
*AQP1*	F = CCTTGGACACCTCCTGGCTATTG	199	56.7	NM_198098.4
	R = CTTCACGCGGTCTGTGAGGT		60.0	
*AQP2*	F = ATGGCGTTTGGCTTGGGTAT	200	60.3	NM_000486.6
	R = GATGTCTGCTGGCGTGATCT		60.2	
*AQP3*	F = ACCAGCTTTTTGTTTCGGGC	111	59.9	NM_004925.5
	R = AGGCTGTGCCTATGAACTGGT		61.5	
*AQP4*	F = CCTCGCTGGTGGCCTTTATGA	207	62.5	NM_001650.7
	R = GTCTTTCCCCTTCTTCTCCTCTCC		61.9	
*AQP5*	F = CCACCTTGTCGGAATCTACTT	205	57.4	NM_001651.4
	R = TTTGATGATGGCCACACGC		59.1	
*AQP6*	F = CCATCATCATTGGGAAGTTCACAG	251	59.8	NM_001652.4
	R = GCGTAGGCTGTTTCACACACTCTC		63.9	
*AQP7*	F = CACAGGCGGTCCACCC	109	59.6	NM_001170.3
	R = TCATGAACTCGGCCAGGAAC		60.0	
*AQP8*	F = ATGTCTGGTCGAACTGCTGG	231	60.0	NM_001169.3
	R = CAGTACGGGAGGAGCATCAC		60.0	
*AQP9*	F = ATCGTGGGAGAAAATGCAAC	196	58.0	NM_020980.5
	R = CAATAATCAGGAGGCCGATG		58.0	
*AQP10*	F = TGCAGTGACAGTGTGCCTAT	178	59.3	NM_080429.3
	R = TGGGTGAGGAGCATGAGTACA		60.3	
*AQP11*	F = TCCGAGTCGACTTGCTCAAA	165	59.3	NM_173039.3
	R = CAGCTCCTGTTAGACTTCCTCC		59.8	
*AQP12a*	F = CCTGCTCTTCCTGCTCTTCC	102	60.1	NM_198998.3
	R = AGAGACTGCTCGGCCATGA		60.7	
*AQP12b*	F = TCTTTGCCACCTTCACCCTC	136	59.9	NM_001102467.2
	R = GTCCTCATCTCCAGGAAGCA		58.8	
*IPO8*	F = GGTGGGGTGTGAGGTAATCC	201	59.7	NM_006390.4
	R = ACTGGTTGAGCTCGTTCTCG		60.0	
*PSMC4*	F = TGGAGGTGCAGGAGGAATACA	162	60.6	NM_006503.4
	R = CTGTGGTAGAGCCCACGATG		60.2	

* expected product size in DNA base pairs; T_M_ = calculated melting temperature in °C.

**Table 3 cancers-15-04507-t003:** Hazard ratios (HR) for patients with Grades 1 and 3 EC show association of overall survival data with *AQP* transcript levels.

Gene	Grade 1 HR (95% Confidence Interval)	Grade 3 HR (95% Confidence Interval)
*AQP1*	HR = 0 (0−Inf)	HR = 1.81 (1.08−3.05)
	logrank P = 0.4 NS	logrank P = 0.024 *
*AQP2*	HR = not determined (>108)	HR = 0.55 (0.3–1.01)
	logrank P = 0.35 NS	logrank P = 0.05 NS
*AQP3*	HR = not determined (>108)	HR = 0.44 (0.41–1.08)
	logrank P = 0.048 NS	logrank P = 0.094 NS
*AQP4*	HR = 0 (0−Inf)	HR = 2.16 (1.33−3.51)
	logrank P = 0.021 NS	logrank P = 0.0014 *
*AQP5*	HR = 0 (0−Inf)	HR = 0.68 (0.4–1.17)
	logrank P = 0.2 NS	logrank P = 0.16 NS
*AQP8*	HR = 0.23 (0.01−3.7)	HR = 1.5 (0.94−2.41)
	logrank P = 0.26 NS	logrank P = 0.09 NS
*AQP11*	HR = not determined (>108)	HR = 1.93 (1.19−3.12)
	logrank P = 0.1 NS	logrank P = 0.0066 *
*AQP12A*	HR = 3.31 (0.21–53.07)	HR = 1.84 (1.13–2.9)
	logrank P = 0.37 NS	logrank P = 0.012 NS
*AQP12B*	HR = 2.8 (0.17–44.86)	HR = 0.75 (0.46–1.23)
	logrank P = 0.45 NS	logrank P = 0.26 NS

* *p* < 0.05; NS not significant.

## Data Availability

Archived datasets generated and analyzed for this study are available on request to one of the corresponding authors.

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
