# Peer review of "Reducing the Invasiveness of Low- and High-Grade Endometrial Cancers in Both Primary Human Cancer Biopsies and Cell Lines by the Inhibition of Aquaporin-1 Channels"

_cancers, 2023, doi:10.3390/cancers15184507_

Round 1

Reviewer 1 Report

The idea of exploring aquaporin channels in terms of cancer and specially endometrial cancers is good. The study provides an overview on the role of AQPs by focusing on the invasiveness of cancer cells as a characteristic. However, I have a few suggestions to improve the study design.

1. The strong conclusions made regarding the role of AQPs in invasiveness need to softened all over the manuscript.

2. I would suggest doing an in vivo study to prove the hypothesis. The effect of these agents in an actual in vivo setting is of utmost importance to prove or disprove the hypothesis.

3. The concentrations of agents used are very high and almost impossible to be achieved in the actual clinical setting, this needs to be discussed.

3. Please provide/specify the loading control blots used in western bot analysis.

Author Response

We are grateful to the reviewers for their insightful analyses of our paper. Their useful feedback has improved the quality of the presentation.

--------

The idea of exploring aquaporin channels in terms of cancer and specially endometrial cancers is good. The study provides an overview on the role of AQPs by focusing on the invasiveness of cancer cells as a characteristic. However, I have a few suggestions to improve the study design.

1. The strong conclusions made regarding the role of AQPs in invasiveness need to softened all over the manuscript.

     We agreed that the conclusions on the potential value of AQP1-targeted treatments needed to be stated more conservatively. The requested changes have been made in the Introduction (Lines 84-86) and Conclusion (717-723) sections.

2. I would suggest doing an in vivo study to prove the hypothesis. The effect of these agents in an actual in vivo setting is of utmost importance to prove or disprove the hypothesis.

     Clarification that in vivo testing is the necessary next step in future research directions has been added to the revised manuscript (lines 86, 701-702, 706-708, 712-723). Testing translation of our in vitro findings to an in vivo system is a compelling but challenging goal that we are keen to pursue. However, before in vivo work can be launched, it is essential to build a database on treatments, doses, durations, and expected outcomes, for which in vitro cell lines and primary cells serve as valuable models. Our work here is the first to establish this database, and is innovative in that the findings are based not only on established cell lines, but also (in partnership with clinical expertise) include parallel testing of primary endometrial cancer cells derived directly from human tissue biopsies. This exciting approach holds merit in setting the stage for an in vivo model.  We fully agree that work in vivo will be the next step, but is outside the scope of this study and will be our main focus for continuing research.  

3. The concentrations of agents used are very high and almost impossible to be achieved in the actual clinical setting, this needs to be discussed.

     In the revised Conclusion, we clarified that the AQP blocking agents tested here should be considered as lead compounds that remain to be optimized (lines 712-714 and 720-723). While the doses do appear high, it is worth noting that both AqB011 and 5-PMFC have been successfully used at the same doses or higher in rodent in vivo models, including work in AqB011-treated mice used to assess the role of AQP1 in cardiac hypertrophy [1], and work with 5-PMFC at doses up to 2.5mM in human red blood cells and rodent models of sickle cell disease for anti-sickling effects [2]. Further work on chemical structure-activity relationships, distribution and metabolism will be needed along with in vivo studies for progress towards clinical translation. This concern has been addressed in the revised MS in discussion section (lines 701-714).

4. Please provide/specify the loading control blots used in western bot analysis.

     This excellent suggestion has been incorporated as requested. The full blot images now include the loading control lanes (alpha-tubulin), and have been added to Figure 8 and the Supplementary Material in Figure S4.

----References cited----

1. Montiel, V.; Bella, R.; Michel, L.Y.M.; Esfahani, H.; De Mulder, D.; Robinson, E.L.; Deglasse, J.P.; Tiburcy, M.; Chow, P.H.; Jonas, J.C.; et al. Inhibition of aquaporin-1 prevents myocardial remodeling by blocking the transmembrane transport of hydrogen peroxide. Sci Transl Med 2020, 12, doi:10.1126/scitranslmed.aay2176.

2. Chow, P.H.; Cox, C.D.; Pei, J.V.; Anabaraonye, N.; Nourmohammadi, S.; Henderson, S.W.; Martinac, B.; Abdulmalik, O.; Yool, A.J. Inhibition of the aquaporin-1 cation conductance by selected furan compounds reduces red blood cell sickling. Frontiers in Pharmacology 2022, 12, 794791.

Reviewer 2 Report

Khan and coauthors conducted original research focused on evaluating the ability of molecules that inhibit Aquaporin-1 channels to reduce the invasiveness of high- and low-grade endometrial cancer cells cultured in vitro. This is the first time that the role of  agents blocking Aquaporin-1 channels has been tested on endometrial cancer cells.

The manuscript is well structured and the study is well conducted

However, there are some aspects that can be improved:

1.       In paragraph  4. Discussion, the authors should better underline the limitations due to the low number of women with endometrial cancers recruited to obtain human EC biopsy collection and primary cell cultures.

2.       Paragraph 5 Conclusions should be entirely rewritten, first emphasizing the limitations inherent in each in vitro model in understanding what really happens in vivo and then proposing the directions in which research in possible clinical applications could evolve in the future. However, the authors should avoid addressing the concept of using AQP1 blocker molecules as a fertility-sparing treatment if they start from the concept of controlling invasion and the progression of disease and emphasize the importance of combining multiple approaches in the treatment of advanced cancer.

  The authors should check the text of their manuscript very carefully to correct many typos that make it difficult to read, especially in paragraphs 2. Materials and methods, 3. Results and 4. Discussion.

Author Response

We are grateful to the reviewers for their insightful analyses of our paper. Their useful feedback has improved the quality of the presentation.

-------

Khan and coauthors conducted original research focused on evaluating the ability of molecules that inhibit Aquaporin-1 channels to reduce the invasiveness of high- and low-grade endometrial cancer cells cultured in vitro. This is the first time that the role of  agents blocking Aquaporin-1 channels has been tested on endometrial cancer cells. The manuscript is well structured and the study is well conducted. However, there are some aspects that can be improved:

1. In  Discussion, the authors should better underline the limitations due to the low number of women with endometrial cancers recruited to obtain human EC biopsy collection and primary cell cultures.

     We agree that the sample size of 8 biopsies should be acknowledged as a limitation, and this is now addressed in the revised manuscript (lines 706-708). However, it is worth pointing out that our inclusion of clinical tissues is a novel contribution in this field, which has relatively few publications with primary endometrial cancer cells. Our sample size of 8 provided statistically significant outcomes, and the findings meshed with the results from the endometrial cancer cell lines, adding parallel lines of supporting evidence. A substantial collaborative effort was required to collect and evaluate 12 human endometrial cancer tissues, four of which had to be excluded due to the dominance of fibroblast growth, as documented in the Methods and Supplementary data. Nonetheless, the reviewer is correct that expanding this sample size will be beneficial for future investigations, and we are pleased to acknowledge this in the revised text (lines 706-708).

2. Conclusions should be entirely rewritten, first emphasizing the limitations inherent in each in vitro model in understanding what really happens in vivo and then proposing the directions in which research in possible clinical applications could evolve in the future. However, the authors should avoid addressing the concept of using AQP1 blocker molecules as a fertility-sparing treatment if they start from the concept of controlling invasion and the progression of disease and emphasize the importance of combining multiple approaches in the treatment of advanced cancer.
     We apologize for the lack of clarity in the original version of this MS. We had listed limitations and suggested future directions, but in retrospect we can see these points were lost by being scattered through the manuscript, and would benefit from being reorganized in one place as requested by the reviewer. The suggested consolidation of ideas has been done in the revised manuscript (lines 701-708, and 712-723).

Comments on the Quality of English Language--- The authors should check the text of their manuscript very carefully to correct many typos that make it difficult to read, especially in paragraphs 2. Materials and methods, 3. Results and 4. Discussion.

     We appreciated the reviewer's suggestion that we check spelling, and have corrected typographic errors in the noted sections of the MS.

Round 2

Reviewer 1 Report

The authors have provided satisfactory justification for the concerns/questions raised.